# Neutrophil Extracellular Traps: A Crucial Factor in Post-Surgical Abdominal Adhesion Formation

**DOI:** 10.3390/cells13110991

**Published:** 2024-06-06

**Authors:** Yuqing Lu, Julia Elrod, Martin Herrmann, Jasmin Knopf, Michael Boettcher

**Affiliations:** 1Department of Pediatric Surgery, University Medical Center Mannheim, University of Heidelberg, 68167 Mannheim, Germany; 2Department of Internal Medicine 3—Rheumatology and Immunology, Friedrich Alexander University Erlangen-Nürnberg (FAU) and Universitätsklinikum Erlangen, 91054 Erlangen, Germany; 3Deutsches Zentrum für Immuntherapie (DZI), Friedrich Alexander University Erlangen-Nürnberg (FAU) and Universitätsklinikum Erlangen, 91054 Erlangen, Germany

**Keywords:** peritoneal adhesion, surgery, neutrophil extracellular traps, DNase1, DNase1L3

## Abstract

Post-surgical abdominal adhesions, although poorly understood, are highly prevalent. The molecular processes underlying their formation remain elusive. This review aims to assess the relationship between neutrophil extracellular traps (NETs) and the generation of postoperative peritoneal adhesions and to discuss methods for mitigating peritoneal adhesions. A keyword or medical subject heading (MeSH) search for all original articles and reviews was performed in PubMed and Google Scholar. It included studies assessing peritoneal adhesion reformation after abdominal surgery from 2003 to 2023. After assessing for eligibility, the selected articles were evaluated using the Critical Appraisal Skills Programme checklist for qualitative research. The search yielded 127 full-text articles for assessment of eligibility, of which 7 studies met our criteria and were subjected to a detailed quality review using the Critical Appraisal Skills Programme (CASP) checklist. The selected studies offer a comprehensive analysis of adhesion pathogenesis with a special focus on the role of neutrophil extracellular traps (NETs) in the development of peritoneal adhesions. Current interventional strategies are examined, including the use of mechanical barriers, advances in regenerative medicine, and targeted molecular therapies. In particular, this review emphasizes the potential of NET-targeted interventions as promising strategies to mitigate postoperative adhesion development. Evidence suggests that in addition to their role in innate defense against infections and autoimmune diseases, NETs also play a crucial role in the formation of peritoneal adhesions after surgery. Therefore, therapeutic strategies that target NETs are emerging as significant considerations for researchers. Continued research is vital to fully elucidate the relationship between NETs and post-surgical adhesion formation to develop effective treatments.

## 1. Background

As many as 93–100% of patients develop adhesions after abdominal surgery, which can result in intestinal obstruction, pelvic pain, and infertility complications [1]. These adhesions may respond naturally to tissue injury during surgery, inflammation, or infection [2,3]. The formation of adhesions is a complex process involving multiple cellular and molecular events, including coagulation, fibrinolysis, inflammation, angiogenesis, and extracellular matrix (ECM) remodeling [3,4]. Neutrophil extracellular traps (NETs) are structures composed of deoxyribonucleic acid (DNA), histones, and granule proteins that are released by neutrophils [5]. NET formation is a unique process in immune defense in humans. NETs were initially discovered as a defense mechanism against microbial infections as they can trap and neutralize pathogens [6]. However, recent research has suggested that NETs can also play a role in non-infectious conditions, including inflammation, autoimmune diseases, and tissue repair [7]. The relationship between NETs and adhesions lies in their involvement in the host response to tissue injury and inflammation [8]. When tissue injury occurs, neutrophils are rapidly recruited to the site of the injury or infection [9]. Upon activation, neutrophils can release NETs to neutralize pathogens; however, NETs can also contribute to tissue damage and inflammation due to cytotoxic components [10,11]. Several studies suggest that the recruitment and activation of neutrophils at the site of injury may contribute to adhesion formation through the release of pro-inflammatory mediators and the promotion of fibroblast activation [12,13,14], raising further questions on their particular significance. Duan Z et al.’s work revealed that measuring the neutrophil extracellular trap formation index (NFI) in postoperative drainage fluid offers a more sensitive and specific method for early prediction of deep surgical site infections compared to traditional serum infection indicators like CRP and PCT [15]. Consequently, the objective of this review was to examine research focusing on the role that neutrophils play in abdominal adhesions. Understanding this relationship could lead to novel therapeutic strategies for preventing or treating postoperative adhesions.

## 2. Review

### 2.1. Search Strategy

Search strategy and selection criteria: This review was performed according to PRISMA (Preferred Reporting Items for Systematic Reviews and Meta-Analyses) guidelines [16]. Electronic databases, including PubMed and Google Scholar, were systematically searched using the keywords or medical subject headings (MeSHs) “Tissue Adhesion” (or “Surgical Adhesion”, “Surgery-Induced Tissue Adhesions”, “Surgical Adhesion”, “Surgery-Induced Tissue Adhesion”) and “Extracellular Traps” (or “Neutrophil Extracellular Traps”, “Extracellular DNA Traps”, “NETs (Neutrophil Extracellular Traps)”). The literature search was conducted with a timeframe from January 2003 to May 2023 and the search was not restricted to specific languages. The following selection process was conducted, according to the PRISMA 2020 checklist [16] (see also Figure 1): In the first step, duplicates were removed. In the next step, studies were screened based on the abstract only by an independent reviewer and finally included if they contained original data relevant to the topic. Next, a full-text article assessment was performed for the remaining articles, and those with an inadequate methodology and insufficient data and those lacking adhesion markers were eliminated according to the Cochrane Risk of Bias Tool [17]. In the last step, a quality assessment was performed for the remaining articles, adhering to the Critical Appraisal Skills Programme (2023) (CASP) qualitative study checklist [18]; see Table 1.

### 2.2. Screening and Quality Assessment

Record identification and removal of duplicates: An initial search conducted in the PubMed and Google Scholar databases yielded 912 records. The removal of duplicates reduced the pool to 709 unique articles. Abstract screening: Subsequent screening of the titles and abstracts led to the exclusion of 582 articles. The exclusion criteria were irrelevance to the research question and non-adherence to the predefined inclusion criteria; see Figure 1. Full-text article eligibility assessment: Further evaluation of the full-text articles resulted in the exclusion of 109 records due to inadequate methodological description. Seven studies were excluded for providing insufficient data, which raised concerns about their reproducibility and reliability. Four studies were excluded because they did not utilize specific markers for adhesion; see Figure 1.

Quality Assessment: A detailed quality assessment was conducted on the remaining seven studies using the Critical Appraisal Skills Programme (CASP) checklist [26]. Overall, most studies conformed to the majority of the checklist criteria; see Table 1.

Results: These 7 studies explore various aspects of NET formation and its implications, illustrating a complex interplay of factors contributing to adhesion formation. One study illustrates how mechanical injury to the mesothelial layer during abdominal surgery leads to the recruitment of neutrophils and monocytes, whose formation of NETs is implicated in the pathogenesis of peritoneal adhesions, suggesting that modulating these immune responses may prevent such occurrences [25]. Additionally, research involving gastric cancer patients post-surgery shows that increased NET production by low-density neutrophils facilitates tumor cell attachment and growth, proposing that DNase treatment to disrupt these NETs could prevent peritoneal recurrence [22]. Moreover, investigations reveal that early-recruited neutrophils release NETs that activate the STING-associated inflammatory response, with interventions that disrupt NETs and inhibit STING signaling markedly reducing adhesion burden, thus identifying NET/STING pathways as potential therapeutic targets [24]. Similarly, evidence from murine models indicates that NETs are crucial in forming peritoneal adhesions, with DNase treatments significantly reducing these adhesions, suggesting potential therapeutic approaches pending further clinical validation [23]. Furthermore, another study demonstrates that mesothelial cells in the peritoneal cavity can undergo a mesothelial-to-mesenchymal transition, contributing to the formation of adhesions, with the blockade of the transforming growth factor beta (TGF-β) pathway offering a novel approach to mitigate these effects [21]. In parallel, research on peritoneal fibrosis highlights the role of inflammation and injury from clinical events like surgery in driving fibrosis through complex interactions among myofibroblasts, leukocytes, and other cell types, contributing to the deterioration of the peritoneal membrane [20]. Lastly, a study on the combined treatment with antithrombin and a PAD4 inhibitor in mice suggests that this approach effectively reduces postoperative adhesion formation by mitigating thrombin and NET-related processes, offering promising insights into managing surgical outcomes [19]. Collectively, these studies not only underscore the multifaceted role of NETs in peritoneal adhesion formation but also open avenues for targeted interventions that could ameliorate or prevent the adverse effects of surgical interventions.

Four main parts will be discussed in more detail in the following paragraphs: the pathogenesis of adhesions, the prevention and treatment of NETs, the potential impact of NET inhibition on adhesion formation, and a feasibility analysis of suppressing NETs in post-surgical adhesion.

### 2.3. Pathogenesis of Adhesions

Adhesions are fibrous bands between internal organs [27]. They form due to events like coagulation, inflammation, and angiogenesis [3,28]. Adhesions can result from surgery, trauma, or inflammatory conditions [28,29]. Upon tissue injury, the body initiates a complex response beginning with the activation of the coagulation cascade. This cascade results in fibrin production, forming a temporary matrix that not only stabilizes the wound but also prevents bleeding [30,31,32]. As a result of injury, inflammatory mediators are released, including cytokines, chemokines, and growth factors, which promote the recruitment of immune cells such as neutrophils and macrophages to the injured area [33,34,35,36]. Another critical element in this process is the mesothelium, a protective layer of cells lining the internal organs and providing a non-adhesive surface [37]. However, when surgical injuries disrupt this layer, there is a significant risk of adhesion formation, highlighting the delicate balance within the body’s internal environment [37]. This disruption leads to a phenomenon known as mesothelial-to-mesenchymal transition (MMT), whereby mesothelial cells acquire a fibroblast-like phenotype. These transformed cells then produce extracellular matrix (ECM) components, further contributing to adhesion formation [37,38,39]. Interestingly, while fibrin deposition acts as a supportive matrix for cell migration and temporary wound stabilization, it also plays a pivotal role in adhesion formation [40,41]. Under normal conditions, this fibrin matrix is gradually degraded by fibrinolytic enzymes like plasmin, which serves to prevent excessive adhesion formation [42,43]. However, an imbalance between fibrin deposition and fibrinolysis can lead to the persistence of fibrin, thereby promoting adhesion [43]. Furthermore, angiogenesis emerges as a critical factor in the development of adhesions. It ensures the supply of oxygen and nutrients to the growing fibrous tissue, underscoring the complexity of adhesion development [44,45]. Angiogenesis is regulated by various pro- and antiangiogenic factors such as vascular endothelial growth factor (VEGF) [46,47] and thrombospondin-1 [48,49,50].

A summary of the pathogenesis, mechanisms, and interventional strategies of adhesion formation can be found in Table 2.

### 2.4. NET Formation and Degradation

In the realm of innate immunity, neutrophil extracellular traps (NETs) serve as a frontline defense mechanism against invading pathogens. These web-like structures are released by activated neutrophils through a dynamic sequence of events, including nuclear decondensation, cytoplasmic content release, and ultimately, cell rupture—a process collectively referred to as NETosis [93,94,95]. Distinct from traditional apoptosis, NETosis is characterized by the explosive release of cellular components that ensnare and neutralize pathogens [96,97,98,99]. The nomenclature “NETosis” merges “NET” with “apoptosis,” highlighting its deviation from conventional programmed cell death mechanisms. Within the spectrum of NETosis, two primary forms are recognized: suicidal and vital NETosis [100]. Suicidal NETosis, or lytic NETosis, primarily involves a cell death program driven by the production of reactive oxygen species (ROS), either through an NADPH oxidase (NOX)-dependent pathway or via mitochondrial-derived ROS (mROS) from a NOX-independent pathway [101]. Intriguingly, suicidal NETosis can also occur without the activation of NOX2, driven instead by extracellular Ca2+ influx, which can be stimulated by agents like fungal ionophores (e.g., nigericin, ionomycin) and granulocyte macrophage colony-stimulating factor (GM-CSF) [3].

In contrast, vital NETosis does not result in neutrophil death. This pathway allows neutrophils to release NETs while remaining alive and functionally active, which is crucial during acute infections where a rapid immune response is necessary. Both mechanisms have been well documented but serve different roles depending on the inflammatory or infectious context, including aseptic conditions as opposed to responses to microbial infections or in autoimmune diseases [100,102].

Specifically, the initiation of NET formation is triggered by various stimuli, including pathogens and immune mediators such as interleukin-8 (IL-8) and lipopolysaccharide (LPS) [103,104,105]. The process involves protein kinase C (PKC) activation and reactive oxygen species (ROS) production via nicotinamide adenine dinucleotide phosphate (NADPH) oxidase [98,106,107]. Subsequently, increased ROS activate enzymes like neutrophil elastase (NE), leading to chromatin decondensation aided by myeloperoxidase (MPO) [108,109]. This change, along with the formation of pores by gasdermin D (GSDMD), results in the formation of NETs [110,111,112].

Peptidyl arginine deiminase 4 (PAD4) converts arginine residues on histones (proteins that help compact DNA) into citrulline, a process known as citrullination [113]. DNA is unfolded or decondensed as a result of this alteration in the charge distribution of histones [114]. As a result, the transcription machinery can access DNA more easily, which in turn affects gene expression [115]. This chromatin decondensation process is essential for NET formation, as it allows for the re-assembling of nuclear components with cytoplasmic granule proteins. Concurrently, GSDMD, a pore-forming protein, is cleaved by caspases to form pores in the plasma membrane [116]. Through these pores, the decondensed chromatin and proteins are ejected, which then form NETs that trap and neutralize pathogens.

However, the degradation of NETs is crucial for resolving inflammation and preventing pathologies such as tissue adhesion [117,118,119]. The primary mechanism by which NETs are degraded involves the action of nucleases, primarily deoxyribonucleases (DNases), which cleave their DNA backbone of NETs [22,119]. Table 3 summarizes the critical factors involved in NET degradation.

### 2.5. The Role of NETs in Adhesion Development

Neutrophils, especially NETs, may be essential during peritoneal adhesion. Recently, it has been suggested that NETs serve as important scaffolds for adhesion formation [8,138]. NETs may affect peritoneal adhesions at various levels. Post-surgical adhesion formation is a complex process involving multiple cell types and signaling pathways [3]. This complexity originates from the consistent recruitment of immune cells [25,139] coupled with the unchecked proliferation of fibroblasts [140,141] and mesothelial cells [142]. Neutrophils play a pivotal role in the early stages of adhesion formation, acting as key mediators in this complex physiological process. Dominant within the initial adhesive environment, neutrophils regulate the release of vital mediators such as IL-8, interleukin-1 beta (IL-1β), and ROS [143]. An essential pathway of note in this context is the ROS signaling pathway. For instance, transmembrane and immunoglobulin domain-containing 1 (TMIGD1) has been shown to thwart abdominal adhesion formation by mitigating oxidative stress in the mitochondria of peritoneal mesothelial cells [144]. Neutrophil extracellular traps (NETs) play an important role in this landscape. The concentration of NETs peaks during the 1–3 day window following surgery [25], and their persistence, extending beyond the lifespan of neutrophils, imparts a prolonged functional importance that persists even after neutrophil apoptosis [145]. However, it is the complex interplay between NETs and various cellular components that emphasizes their collective importance in the postoperative milieu. Central to this interaction is the concept of pathway interdependence, which accentuates the synergistic relationships between cellular processes and mediators in the development of post-surgical adhesions. For example, the ROS interplay with NET formation is diverse, influencing whether NET formation manifests beneficial or detrimental effects. Excessive ROS, produced during neutrophil activation, are known to trigger NET formation by causing severe DNA damage, like oxidizing guanine to 8-oxo guanine [146]. This eventually initiates a DNA repair pathway, resulting in chromatin decondensation.

In the context of peritoneal adhesions, NET formation predominantly occurs within the peritoneal cavity, particularly proximal to post-capillary venules and at locations affected by surgical interventions or tissue injuries [5]. This spatial distribution is especially pertinent during abdominal surgeries, where neutrophils are stimulated to release NETs in reaction to tissue damage and the presence of foreign materials such as surgical sutures. The formation of NETs at these sites initiates a pro-inflammatory cascade that facilitates critical processes like fibrin deposition and fibroblast activation, which are essential for the development of peritoneal adhesions [147]. Furthermore, the process of histone citrullination, a modification associated with NET formation, serves as a precise biomarker for this phenomenon [5]. What is more, the level of H3cit (citrullinated histone H3) was developed as a biomarker for some diseases associated with NETs, like arterial thromboembolism (ATE) [148], tumors [115], and abdominal aortic aneurysms [149]. The quantification of histone citrullination offers a direct measure of NET activity, thereby linking NET formation to the progression of surgical adhesions. This biomarker provides an invaluable metric for both researching the underlying mechanisms of adhesion formation and evaluating clinical interventions aimed at modulating NET formation to mitigate its pathological effects.

After abdominal surgery, neutrophils are activated to produce NETs through a complex interplay of physiological and biochemical processes. Surgical trauma and tissue damage result in the release of damage-associated molecular patterns (DAMPs), which are recognized by pattern recognition receptors (PRRs) on neutrophils, initiating their activation. The subsequent inflammatory response is characterized by the release of cytokines (e.g., IL-1β, IL-6, TNF-α) and chemokines (e.g., IL-8, CXCL1) [100,150], which prime neutrophils and enhance their sensitivity to stimuli that induce NET formation. Additionally, the potential for contamination or infection during surgery introduces pathogen-associated molecular patterns (PAMPs) that further activate neutrophils through PRRs [151]. Activated platelets, a common occurrence due to tissue injury and bleeding, release mediators such as platelet factor 4 (PF4) and CD40L, and directly interact with neutrophils via surface molecules, thereby promoting NET formation [152]. ROS produced by neutrophils upon activation, alongside intracellular signaling pathways involving enzymes like NADPH oxidase and MPO, contribute to chromatin decondensation and the active release of NETs through NET formation [10]. And many new triggers have been discovered, including those involved in sterile inflammation [153]. Moreover, a recent study by Pandolfi, Laura et al. suggests that neutrophil extracellular traps (NETs), induced by SARS-CoV-2 and combined with factors secreted by alveolar macrophages, can drive the epithelial–mesenchymal transition (EMT) in lung epithelial cells, potentially leading to lung fibrosis in severe COVID-19 patients [154]. Since epithelial cells are also involved in adhesion formation, EMT induced by NET formation might similarly contribute to adhesion development.

### 2.6. The Potential Impact of NET Inhibition on Adhesion Formation

Understanding the inhibition of NETs is crucial for developing therapeutic strategies to modulate their formation in pathological conditions, where excessive or dysregulated NET formation contributes to disease progression [155]. Various inhibitors and molecular targets that can effectively inhibit NET formation have been identified. In brief, NETs and adhesions are implicated in the inflammatory and healing responses to tissue injury; NETs, released by neutrophils, can induce fibroblast activity and thrombosis [156], thus contributing to adhesion between organs. The dysregulation of NET formation or removal plays a critical role in abnormal tissue repair, potentially exacerbating adhesion development [157]. A summary of the role of NETs in peritoneal adhesion formation is summarized in Table 4.

### 2.7. Feasibility Analysis of Suppressing NETs in Post-Surgical Adhesion

It is essential to note that the choice of inhibitor and its efficacy may depend on the context of a specific disease or inflammatory condition. NETs are a component of the complex immune response during tissue injury and inflammation. Therefore, the effect of inhibiting NETs on adhesion formation may be context-dependent and varies based on specific surgical or pathological conditions. More research is required to directly investigate the relationship between NET inhibition and adhesion formation and to determine whether targeting NETs could be a viable strategy for adhesion prevention while minimizing the adverse effects.

The role of NETs (neutrophil extracellular traps) in wound healing, especially within the context of bowel anastomosis, remains an underexplored area of research. However, emerging studies underscore a significant interplay between NETs and the wound healing process [157]. A pivotal study published in *Nature Medicine* in 2015 by Wong et al. illustrated that the presence of diabetes increases the formation of NETs within wounds, prolonging the healing process in diabetic wounds [174]. Concurrently, another investigation revealed that during lung ischemia–reperfusion, the release of mitochondrial DNA (mtDNA) induces the formation of TLR9-mediated NETs, intensifying lung injury [175]. Given that injuries from bowel anastomosis are essentially ischemic reperfusion injuries or arise from tissue hypoxia [176], it is possible that the generation of NETs has a negative impact on the repair of tissues affected by ischemia and hypoxia. Contrarily, in vitro analyses have indicated that under suboptimal concentrations, NETs might bolster the proliferation of keratinocytes, driven by the activation of the nuclear factor kappa-light-chain-enhancer of activated B cells (NF-κB) pathway [177]. This suggests a concentration-dependent regulatory role of NETs in wound healing. This observation aligns with the findings of Saffarzadeh et al. which elucidated that the cytotoxic effects of NETs on epithelial and endothelial cells are predominantly determined by concentration levels [178]. Interestingly, this cytotoxicity is not modulated by DNA fragments but is orchestrated by histones and MPO, while NE does not confer any cytotoxic effects. Histones, as primary constituents of NETs, act as damage-associated molecular patterns (DAMPs) and are hypothesized to be cytotoxic to epithelial and endothelial cells. Additionally, proteins released in conjunction with NETs, such as NE and matrix metalloproteinase (MMP), might impede wound healing by detrimentally affecting the ECM [179]. In summation, while the antimicrobial attributes of NETs offer a positive influence on wound healing, their deleterious effects, especially in conditions like diabetes, are pronounced and frequently correlate with hyperactive NET formation. These findings are summarized in Table 5.

NETS exhibit a bifunctional role in both wound healing and broader physiological responses, illustrating the multifaceted nature of their biological effects. Excessive NET production may precipitate augmented adhesion formation, suggesting that their inhibition could be beneficial. Conversely, it is conceivable that NET suppression might impede wound healing, for instance, in the context of postoperative recovery. Within this complex interplay, the process of anastomosis stands out as a critical determinant of surgical recuperation and sustained patient prognosis. Anastomosis, entailing the surgical juncture of two bodily passages, is essential for re-establishing physiological continuity post-surgery. Anastomotic insufficiencies pose a critical concern in surgical outcomes, indicating the significance of intricate biological processes beyond the well-recognized ischemia–reperfusion injury [196]. The intricate interplay of the gut microbiome in surgical recovery is emerging as a critical factor [197]. This complex network of commensal bacteria, predominantly housed in the colon’s mucus layer, plays a pivotal role in various metabolic and immune processes [198]. Perioperative interventions, such as bowel preparation, antibiotic administration, fasting, and the stress of surgery itself, can severely disrupt this delicate microbial balance [199]. Such disruptions, known as dysbiosis, may lead to nutrient depletion, diminished microbial diversity, and consequently, compromised anastomotic healing due to increased susceptibility to pathogenic bacteria [200]. Moreover, an altered gut microbiome can promote NET formation, a hypothesis supported by current studies [201,202,203]. This is mediated through various mechanisms: microbial metabolites like short-chain fatty acids (SCFAs) modulate immune cell activity [204,205]; dysbiosis increases intestinal permeability, enhancing exposure to pro-inflammatory stimuli like lipopolysaccharides (LPSs) that trigger NET release [206,207,208]; and pathogenic microbes can directly stimulate NET formation via interactions with neutrophil pattern recognition receptors (PRRs) [209].

For these reasons, a cautious approach is warranted. Selective inhibition targeting modulators that minimally affect granule proteins, such as myeloperoxidase (MPO) or neutrophil elastase (NE), may represent a judicious strategy. Previous research highlights the role of neutrophil extracellular traps (NETs) in providing a scaffold for adhesion formation, with DNase administration shown to disrupt this process effectively [23]. Consequently, therapeutic applications of DNases could offer a viable intervention to prevent peritoneal adhesion without hindering the normal healing process.

## 3. Conclusions

Throughout this review, the involvement of neutrophil extracellular traps (NETs) in the development of postoperative abdominal adhesions has been critically examined. Emerging evidence suggests that interventions targeting NETs could represent a novel therapeutic pathway for the prevention of adhesions. However, the potential effect of NET inhibitors on essential healing processes necessitates further in-depth research. Future studies must delineate NETs’ precise role in the formation of adhesions and rigorously evaluate the therapeutic effectiveness and safety of NET-targeted therapies. Advancing our understanding of NET-mediated adhesion could lead to innovative strategies that reduce surgical adhesion-related complications.

## Figures and Tables

**Figure 1 cells-13-00991-f001:**
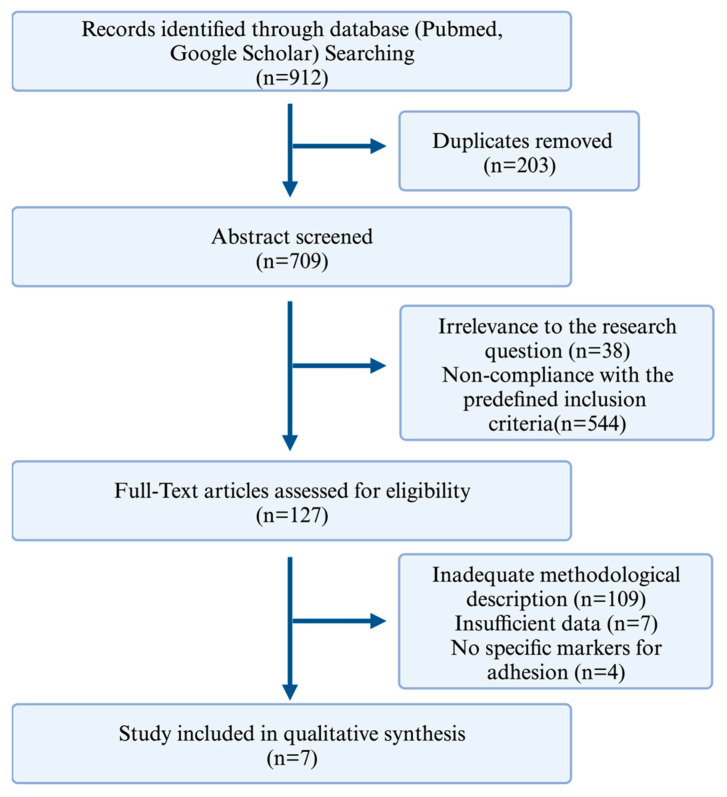
A PRISMA [16] flow diagram showing the search and selection process. The initial search across PubMed and Google Scholar databases yielded 912 records. After the removal of duplicates, this was refined to 709 unique articles. A review of titles and abstracts led to the exclusion of 582 articles due to irrelevance or those not meeting the inclusion criteria. A total of 127 records underwent full-text article assessment for eligibility and a quality assessment, finally yielding 7 articles for inclusion in our final review.

**Table 1 cells-13-00991-t001:** Quality assessment using Critical Appraisal Skills Programme (2023) (CASP) qualitative study checklist.

	a	b	c	d	e	f	g	h	i	j
**Sudo, Makoto et al. [19]**	Y	Y	Y	Y	Y	Y	Y	Y	Y	Y
**Terri, Michela et al. [20]**	Y	Y	Y	Y	Y	Y	Y	Y	Y	Y
**Sandoval, Pilar et al. [21]**	Y	N	Y	Y	N	Y	Y	Y	Y	Y
**Kanamaru, Rihito et al. [22]**	Y	Y	Y	Y	Y	Y	Y	Y	Y	Y
**Elrod, Julia et al. [23]**	Y	Y	Y	Y	Y	Cannot tell	Y	Y	Y	Y
**Hu, Qiongyu et al. [24]**	Y	Y	Y	Y	Y	N	Y	Y	Y	Y
**Jonathan M. Tsai et al. [25]**	Y	Y	Y	Y	Y	Y	Y	Y	Y	Y

Note: (a) Was there a clear statement of the aims of the research? (b) Is a qualitative methodology appropriate? (c) Was the research design appropriate to address the aims of the research? (d) Was the recruitment strategy appropriate to the aims of the research? (e) Was the data collected in a way that addressed the research issue? (f) Has the relationship between researcher and participants been adequately considered? (g) Have ethical issues been taken into consideration? (h) Was the data analysis efficiently rigorous? (i) Is there a clear statement of findings? (j) How valuable is the research? Y = yes; N = no.

**Table 2 cells-13-00991-t002:** Overview of adhesion pathogenesis, mechanisms, and interventional strategies. ECM—extracellular matrix, VEGF—vascular endothelial growth factor, siRNA—small interfering RNA, and shRNA—short hairpin RNA.

Overview of Adhesion Pathogenesis, Mechanisms, and Interventional Strategies		
Aspect	Description	References
Etiology and Pathogenesis	Adhesions, fibrous bands between organs, result from coagulation, inflammation, and angiogenesis initiated by tissue injury from surgery, trauma, or inflammation, leading to fibrin deposition and cytokine-mediated immune activation.	[3,25,27,28,29,30,31,32,33,35]
Fibrin Deposition and Fibrinolysis	Fibrin forms a matrix for cell adhesion; imbalance due to impaired fibrinolysis promotes adhesion, as seen with reduced plasmin activity.	[40,41,42,43,51]
ECM Remodeling and Fibroblast Proliferation	Fibroblasts migrate to the injury site, expanding and remodeling ECM with collagen, fibronectin, elastin, proteoglycans, and glycoproteins, essential for adhesion maturation.	[52,53,54,55,56]
Angiogenesis	Neovascularization, essential for supplying nutrients to fibrotic tissue, is driven by factors like VEGF and inhibited by thrombospondin-1.	[44,45,46,47,48,49,50]
Mesothelial Cell Dysfunction	Surgical trauma induces mesothelial-to-mesenchymal transition, with mesothelial cells producing ECM, facilitated by disruptions to the protective mesothelial layer.	[37,38,39,57,58]
Prophylactic Surgical Techniques	Precision techniques, including minimally invasive surgery like laparoscopy, aim to minimize tissue trauma and reduce foreign material use to prevent adhesions.	[59,60,61,62]
Mechanical Barrier Interventions	Bioresorbable films and gels, such as Seprafilm (sodium hyaluronate and carboxymethylcellulose), and hydrogels, such as polyethylene glycol-based products, are used to prevent tissue adhesion.	[63,64,65,66,67,68,69]
Pharmacological Interventions	Anti-inflammatory drugs, corticosteroids, tissue plasminogen activators, and antiangiogenic agents are investigated to mitigate adhesion formation.	[70,71,72,73,74]
Regenerative Medical Approaches	Stem cell therapy, specifically mesenchymal stem cells, and tissue engineering with bioactive scaffolds or hydrogels are explored for their anti-adhesive effects.	[75,76,77,78,79,80]
Targeted Molecular Therapies	Molecular interventions including siRNA or shRNA for gene silencing and monoclonal antibodies targeting specific growth factors or signaling pathways are developed to inhibit adhesion pathways.	[81,82,83,84,85,86,87]
Barrier Implementation and Pharmacological Strategies	Mechanical barriers, such as Seprafilm and pharmacological agents, are utilized during surgery to reduce adhesion incidence, with an increasing interest in their combined roles.	[88,89,90]
Integrated Strategy and Future Research	Appropriate surgical techniques are currently the most effective prevention, with ongoing research into optimizing barrier methods, pharmacological agents, and novel molecular and regenerative therapies.	[59,60,61,62,68,91,92]

**Table 3 cells-13-00991-t003:** Main factors of NET degradation. SLE—systemic lupus erythematosus, DNase1—deoxyribonuclease-1, DNase1L3—deoxyribonuclease-1-like 3, DNase2—Deoxyribonuclease-2, HUVS—hypocomplementemic urticarial vasculitis syndrome, ALI—acute lung injury, ARDS—acute respiratory distress syndrome, DCs—dendritic cells, and MerTK—Mer tyrosine kinase.

Enzyme/Cell Type	Detailed Function and Location	Disease/Condition Association	NET Degradation and Clearance Mechanism	References
DNase1	Cleaves extracellular DNA within NETs; crucial for NET disassembly. Normally active in the circulation.	SLE: Patients experience NET accumulation due to absent or impaired DNase1 activity.	Central to the degradation process of NETs; absence leads to severe accumulation of NETs, highlighting its importance for immune homeostasis.	[120,121,122]
DNase1L3	Shares structural and functional properties with DNase1; crucial for extracellular NET degradation.	Linked to autoimmune diseases due to DNase1L3 deficiency. Specific mutations related to HUVS and SLE.	Similar to DNase1 in preventing an autoimmune response through degradation of NETs	[36,123,124]
DNase2	Functions at an acidic pH within lysosomes; not a primary actor in NET degradation.	Cystic fibrosis: NETs contribute to airway obstruction; DNase2 may degrade DNA in slightly acidic environments.	Less effective than DNase1/L3 at degrading NETs, but may act if airway surface pH is sufficiently acidic, indicating a secondary role in NET clearance.	[125,126,127,128]
Macrophages, phagocytes, and DCs	Phagocytes that engulf and digest NETs through their endosomal–lysosomal system. Express receptors like MerTK for NET recognition. DCs also remove and degrade NETs.	ALI, ARDS, and autoimmune responses: NETs exacerbate inflammation.	Both cell types play crucial roles in physical engulfment and biochemical degradation of NETs, including the production of DNases.	[129,130,131,132,133,134]
Complement-mediated degradation	Part of the innate immune system that can recognize and clear NETs once complement proteins have been deposited.	SLE: NETs act as potent activators of the complement system, impacting SLE pathology.	The complement system marks NETs for phagocytic destruction, aiding in clearance and potentially affecting autoimmune disease progression.	[10,135,136,137]

**Table 4 cells-13-00991-t004:** The role of neutrophil extracellular traps in peritoneal adhesion formation. ECM—extracellular matrix, ROS—reactive oxygen species, PM—polymyositis, ILD—interstitial lung disease, TLR9—Toll-like receptor 9, LF—lung fibroblast, MF—myofibroblast, and TGF-β—transforming growth factor beta.

The Role of Neutrophil Extracellular Traps in Peritoneal Adhesion Formation		
Aspect	Description	References
Inflammatory Response	Inflammation activates cells like neutrophils to release NETs, promoting fibroblast activation, ECM remodeling, and angiogenesis, which are involved in adhesion formation. NETs and adhesions are interconnected through inflammation and tissue injury. Excessive ROS trigger NET formation through severe DNA damage, initiating DNA repair pathways.	[146,158,159,160,161,162,163,164,165,166]
Tissue Injury Response	Adhesions and NETs play significant roles in responding to tissue injury and inflammation. Adhesions form fibrous bands between organs post-injury, while NETs are immune response structures released by activated neutrophils.	[94,167,168,169]
Fibroblast Interaction	NETs interact with fibroblasts, promoting their activation and ECM production, contributing to adhesion formation. They also influence polymyositis (PM)-associated interstitial lung disease (ILD) and the TLR9-miR-7-Smad2 signaling pathway in lung fibroblast (LF) proliferation and myofibroblast (MF) differentiation.	[170,171]
Tissue Remodeling	NETs modulate tissue remodeling and repair, essential for wound healing. Dysregulated NET formation may result in abnormal tissue repair and adhesion formation.	[157,172,173]
Coagulation	NETs interact with the coagulation system, enhancing thrombosis, which contributes to adhesion development by promoting fibrin deposition as a scaffold. Neutrophil-mediated inflammation enhances TGF-β signaling, leading to fibrotic thrombus remodeling.	[156]

**Table 5 cells-13-00991-t005:** The potential therapeutic targets of NET inhibition in the context of adhesion formation. NETs—neutrophil extracellular traps; ROS—reactive oxygen species; NADPH—nicotinamide adenine dinucleotide phosphate; PKC—protein kinase C; NE—neutrophil elastase; MPO—myeloperoxidase; PAD4—peptidylarginine deiminase 4; NSA—necrosulfonamide; GSDMD—gasdermin D; DPI—diphenyleneiodonium; NOX—NADPH oxidase.

The Potential Therapeutic Targets of NET Inhibition in the Context of Adhesion Formation				
Therapeutic Targets	Specific Agents or Approaches	Description and Impact	Examples	References
Inhibition of NET formation	ROS inhibitors	Agents such as DPI and apocynin target the NADPH oxidase complex to reduce oxidative stress and subsequent NET formation.	DPI, apocynin, LDC7559, and NA-11	[111,112,180,181]
	PKC inhibitors	Gö6976 and Ro-31-8220 suppress the protein kinase C pathway crucial in the signaling cascade that leads to NET formation.	Gö6976 and Ro-31-8220	[113,182]
	NE and MPO inhibitors	Sivelestat and ABAH inhibit NE and MPO, respectively, preventing chromatin decondensation critical for NET release.	sivelestat/ONO 5046 and ABAH	[114,115]
	PAD4 inhibitors	Compounds such as Cl-amidine and GSK484 target PAD4, thus blocking histone citrullination, a key step in NET assembly.	Cl-amidine, GSK484, and BMS-P5	[116,117,183]
	GSDMD inhibitors	Blocking of the pore-forming protein gasdermin D hinders the expansion of chromatin and granular proteins during NET formation.	NSA and disulfiram	[118,119,184,185,186,187,188]
Pharmacological agents	NET suppression	Exhibition of anti-inflammatory effects that indirectly reduce NET formation, with chloroquine specifically inhibiting NADPH oxidase and metformin, affecting PKC-βII in neutrophils.	chloroquine, simvastatin, and metformin	[120,189,190,191]
Endogenous molecules	Acceleration of NET degradation	Degradation of extracellular NET structures by endogenous enzymes, potentially reducing NET-mediated thrombosis and improving ischemia–reperfusion outcomes; Pulmozyme® (recombinant human DNase1) has shown efficacy in disintegrating NETs in clinical settings, suggesting its utility in managing NET-related complications.	DNase1 and DNase1L3	[121,122,123,124,192,193,194,195]

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
