# Peer review of "Neutrophil Extracellular Traps: A Crucial Factor in Post-Surgical Abdominal Adhesion Formation"

_cells, 2024, doi:10.3390/cells13110991_

Round 1

Reviewer 1 Report

Comments and Suggestions for Authors

The review paper is entitled "Neutrophil Extracellular Traps: A Crucial Factor in Post-Surgical Abdominal Adhesion Formation". However, after a screening in the literature for studies involving NETs and abdominal adhesion formation, only 7 studies were found and are presented in Table 1. The seven papers presented are referred to as articles 22 to 28 and none of them agree with the list of References at the end of the article. It seems to me that the articles cited as 23 and 25 refer to the same work, which should be number 47. The paper listed as 26 in table 1 appears to be 193 in the reference list and this reference is cited incompletely (without the year, journal published, etc.). In addition, the articles cited as 10 and 193 refer to the same work, but with different authors (check and clarify). As this citation of the 7 articles is confused, it is not clear what their findings/contributions are to the subject of the review. They must be cited and discussed in the text. Are they all original articles or are there review articles among them? In any case, the few findings in the literature do not support NETs as a crucial factor, but rather as a factor that can contribute to this process. I believe that in order to better support this hypothesis, the authors could include the following articles in the review: 1) Dölling M, Herrmann M, Boettcher M. Children (Basel). 2024;11(3):295. doi: 10.3390/children1103029; 2) Kanamaru R. et al. Sci Rep. 2018;8(1):632. doi: 10.1038/s41598-017-19091-2; 3) Duan Z. et al. Ann Transl Med. 2021;9(17):1373. doi: 10.21037/atm-21-1078. A detailed review of all the literature cited in the text is necessary. For example, reference 5 cited in the body of the text (Background, p. 2, line 50) is not related to the topic presented in the sentence. In short, the authors need to review and reflect on the title of the paper and better adapt its background.

Author Response

please see uploaded letter

Reviewer 2 Report

Comments and Suggestions for Authors

The review by Lu et. al., is focused on the role of NETs in development and treatment for abdominal adhesions. A thorough review of the literature was undertaken guided by keywords and subject headings. Table 1 which evaluates the quality of the search strategy is devoid of much useful information. Paragraph 2.1 on the search strategy would be more informative if it evaluated whether the criteria actually served the purpose to identify publications on the relationship between NETs and surgical adhesions. It seems that beginning with 912 records and ending up with only seven studies that were included as the backbone of this review would motivate going back and adjusting the keywords and criteria. There is a dearth of information provided on how neutrophils are activated to produce NETs that may promote the formation of peritoneal adhesions after surgery. This as opposed to what is known how NETs are formed in aseptic conditions versus in response to bacterial, viral, fungal, or parasitic infections, and autoimmune diseases. Unfortunately, there is no discussion of the two different mechanisms that have been identified and well published on, including suicide lytic NETosis versus live cell or vital NETosis. 

In addition, it would be useful to discuss where NETosis occurs in the context of the peritoneal adhesions; is this just proximal to post capillary venules and what is the most accurate measure of NETosis to relate to the adhesions- histone citrullination?

Comments on the Quality of English Language

Quality of language is mostly good.

Author Response

please see uploaded letter

Round 2

Reviewer 1 Report

Comments and Suggestions for Authors

The authors have addressed all my questions and concerns.

Author Response

Thanks.

Reviewer 2 Report

Comments and Suggestions for Authors

This is an acceptable response. One important point made is that a key factor in the formation of adhesions is mesothelial-to-mesenchymal transition. A recent manuscript reports that NET formation is involved in epithelial-mesenchymal transition of lung epithelial cells in progression of COVID-19. This could be discussed in the  mechanism section and also provide impetus to examine therapies to dampen NETs in peripheral tissue.

Author Response

please see upload
